# An Online Non-Invasive Condition Assessment Method of Outdoor High-Voltage SF6 Circuit Breaker

Clailton Leopoldo Da Silva [1,2], Osmar Reis [1], Frederico de Oliveira Assuncao [1], Julio Cezar Oliveira Castioni [1,2], Rafael Martins [2], Carlos Eduardo Xavier [2], Isac Antonio dos-Santos Areias [1], Germano Lambert-Torres [1,*], Erik Leandro Bonaldi [1], Luiz Eduardo Borges-da-Silva [1] and Levy Ely Lacerda Oliveira [1]

[1] Gnarus Institute, Itajuba 37500-052, MG, Brazil
[2] COPEL Geração e Transmissão S.A., Rua José Izidoro Biazetto, 158, Mossunguê, Curitiba 81200-240, PR, Brazil
[*] Correspondence: germanoltorres@gmail.com

**Abstract:** Online monitoring of outdoor high voltage SF6 circuit breakers (HVCBs) is essential to detecting potential damages. To this end, the study of accurate and non-invasive monitoring methods has been significantly investigated in recent decades. Considering that HVCB vibration patterns carry important information about mechanical and electrical integrity and that vibration analysis requires a low level of invasiveness, this article presents methods of obtaining mechanism and reaction times using interference signals of outdoor high voltage SF6 circuit breakers (HVCBs). Compared to traditional methods of monitoring outdoor SF6 circuit breakers that are based on encoders, the proposed method presented ease of installation, as it only requires the insertion of accelerometers. The method of obtaining the mechanism time is based on the use of interference and auxiliary contact signals, in the time domain, where the accelerometer is installed, in the structure of the HVCB, makes it possible to guarantee the moment of the trip command. To obtain the reaction time of each HVCB pole, the envelope technique was applied with the Hilbert transform module of the hearing signal, filtered in a certain resonance band. The proof of the technique of analyzing the vibration of the signal in time was developed with laboratory tests of an HVCB, instrumented with accelerometers and angular encoders. The results obtained via vibration analysis were compared with those obtained via angular encoders and it was concluded that with the acceleration signals, in time, it is possible to obtain performance parameters of an HVCB from its displacement curve. Finally, the online monitoring of the circuit breaker applied in the field is presented, where the acquisition of trip current signals, the condition of the SF6 gas and extinguishing current signals were added to the instrumentation.

**Keywords:** high voltage circuit breaker (HVCB); vibration analysis; timing characteristics; online monitoring

## 1. Introduction

High Voltage Circuit Breakers (HVCBs) are commonly used in the power system. They perform the function of interrupting currents and protecting other network equipment. Their reliability plays an important role in ensuring the safety and stability of the electrical system operation [1–3].

According to the survey on failures and defects in HVCBs carried out by the Conseil International des Grands Resaux Electriquies (CIGRE), 83 utilities from 26 countries claim that most problems are of mechanical origin [4]. Due to the importance of studying mechanical failures in high voltage circuit breakers, there are currently several studies on fault identification in high voltage circuit breakers using vibration, ultrasound, contact displacement and control coil current signals [5,6].

In the analysis of vibrations applied to the HVCB, small changes in its mechanical characteristics, between wear and tear or deformation, have an impact on the variation in

signatures in the respective signals. Regarding the cost of applying a monitoring system based on vibration analysis, it is more advantageous compared to more traditional methods, such as monitoring the displacement of contacts by encoders installed in the HVCB mechanism, which demands a high level of invasiveness in the installation process [7]. Furthermore, the practical application of vibration analysis can significantly increase the reliability of the entire system [8,9]. Each operation of an HVCB occurs in periods that do not exceed 100 ms, where all the energy stored in the mechanism is released for the displacement of the moving contacts [10]. In this sense, a high impact is generated, reflected in high levels of vibration throughout the structure of the HVCB. Possible changes in vibration signals collected during movements can point to potential failures such as insufficient lubrication, slowness in relays and locks, malfunctioning of the damper, etc. [11].

Research involving vibration analysis applied in HVBC can be classified according to the method of extracting characteristics of the signals, such as the domain of analysis: time domain, frequency domain and time–frequency domain. In this work, the signals will be evaluated in the time domain. In this domain, most of the tools found for monitoring HVBC originate from works that seek to extract features in vibroacoustic signals.

Among the applied tools, there is a cross correlation between two signals [12], dynamic time warp algorithm [13,14] and statistical methods such as root mean square, peak value and Kurtosis [15]. For methods based exclusively on the frequency domain, there are works on Zoom FFT (ZFFT) and shirp z transform (CZT) [16]. In works involving the time–frequency domain, there is the application of the Short Time Fourier Transform (STFT) [17], Empirical Mode Decomposition (EMD) [11] and Continuous Wavelet transform (CWT) [18].

In terms of circuit breaker monitoring, current technologies mainly concern condition parameters related to interrupting current, such as fault clearing time, command relay tripping time, breaker tripping time and interrupting service, because no additional sensors are implemented [12]. There is also the analysis of trip coil current signatures that seeks to correlate electrical signatures of the trip coil with the mechanical condition of the circuit breakers [19,20]. The described method of trip coil current analysis in assessing the condition of HVCB can be applied to certain types of circuit breakers (coil spring mechanism); however, it may not be applicable to other types, such as those with hydraulic operating mechanism. The evaluation of mechanical conditions of HVCB of this type is from displacement sensors, auxiliary contacts or, preferably, from vibration signals generated by circuit breaker operations [21].

Compared to the aforementioned methods, the proposed work performs the analysis in the time domain, as in [10,11,13], which provides greater simplicity, as it is a direct analysis, without transformations, and of less complexity for the interpretation of the results; they are only one-dimensional arrays. This work, in comparison with the aforementioned time-domain monitoring techniques, has the advantage of not requiring the implementation of statistical tools, which reduces possible subjectivities in the interpretation of results, considering that it is the most direct method. Consequently, an algorithm for its automation becomes less complex.

This article presents a proposal for an online monitoring system that does not require the installation of encoders, but instead uses tools developed based on vibration analysis to detect events in the travel curve of an HVCB. For this, an HVCB test laboratory was built, whose instrumentation involves encoders and accelerometers, with the objective of evaluating the results, using analysis of the vibrations of the signals in time and the travel curve. These data are obtained via the installed encoders. In Section 2, vibration analysis and its application in circuit breakers will be presented, the contact displacement curve of an HVCB and the respective events will be presented. Section 3 describes the devices used in the tests. Section 4 reports laboratory test results. Section 5 describes the field installation of the monitoring system. Finally, the discussions and conclusions are presented in Section 6.

## 2. Vibration Analysis

The main mission in vibration analysis is to detect a change between a characteristic of a vibration signal, collected during operation and the reference. Typical problems indicated by deviation in vibration pattern include [22]:

- Over-travel in connecting rids;
- Strain of a drive shaft;
- Release of a contact or frame or mobile mechanism;
- Hydraulic/spring mechanism defect.

In an HVCB opening maneuver, the main contacts open first, then the arcing contact opens with the least amount of time. Therefore, measuring the time between the opening of the main and arc contacts would indicate the degree of ablation of the arc contacts.

The circuit breaker's arcing contacts get progressively shorter as they are worn during each operation, [14,23]. So, for an older circuit breaker, as the arcing contact gets shorter, there will be a delay in the instant of arc contact. Therefore, the time difference would be reduced. In practice, the nominal contact instant would remain practically constant, while the arcing contact instant would be delayed during a closing operation [24].

*Event Detection on the Travel Curve via Acceleration Signals*

In this work, Alstom's HVCB GL312 - GE Grid Solution, was maneuvered in opening–closing (OC) events. In each maneuver, the following signals were obtained:

- Vibration signals from two accelerometers (installed on the base of two poles of the circuit breaker);
- Trip current signal;
- Breaker auxiliary contact signal;
- Breaker contact displacement signals obtained via two incremental rotary encoders.

It should be noted that the accelerometers were installed at the base of each circuit breaker pole, as for field installation, it is a safe location, without risk of electric shock, and does not require the circuit breaker to be de-energized.

Thus, with the obtained signals, the events associated with the maneuvers will be calculated with the vibration signals and compared with the reference signal, which are provided via the incremental encoders. Currently, temporal information, such as reaction time and trip instant (trip time) and contact displacement speed information, is not monitored in circuit breakers installed in the field, rather, only in acceptance tests carried out by the manufacturer. Even in predictive maintenance, reaction times and displacement speed are not collected, only the moment of closing or opening of contact and tripping, obtained via electrical signals injected into the pole terminals themselves. The circuit breaker operating mechanism is responsible for moving the contacts when they are requested and the increase in the duration of this displacement leads to a reduction in the useful life of the contacts, as they will be exposed to a longer duration arc. Thus, online monitoring of these mechanical operation times becomes important for the diagnosis of circuit breakers, especially concerning the threat to the reliability and stability of the energy supply that may occur due to its eventual failure [25].

Figure 1 shows a contact displacement curve (travel curve), obtained in an HVCB test laboratory, which will be described in the next section. This curve was obtained when performing an OCO maneuver, yielding three signals:

In the contact displacement curve (travel curve), operating times are obtained, as illustrated in Figure 1; between the times, there are the following:

- Tr-c (Closing reaction time): Time it takes after the trip signal (close command) for the contacts to start moving;
- Tr-o (Opening reaction time): Time it takes after the trip signal (open command) for the contacts to start moving;

- Close command: Instant when the command for operating the circuit breaker (trip) was issued;
- To-c (Mechanism closing time): This represents the interval between the closing command signal and the closing of the main contact;
- To-o (Mechanism opening time): This represents the interval between the opening command signal and the opening of the main contact;
- Highlights: Breaker speed calculation points.

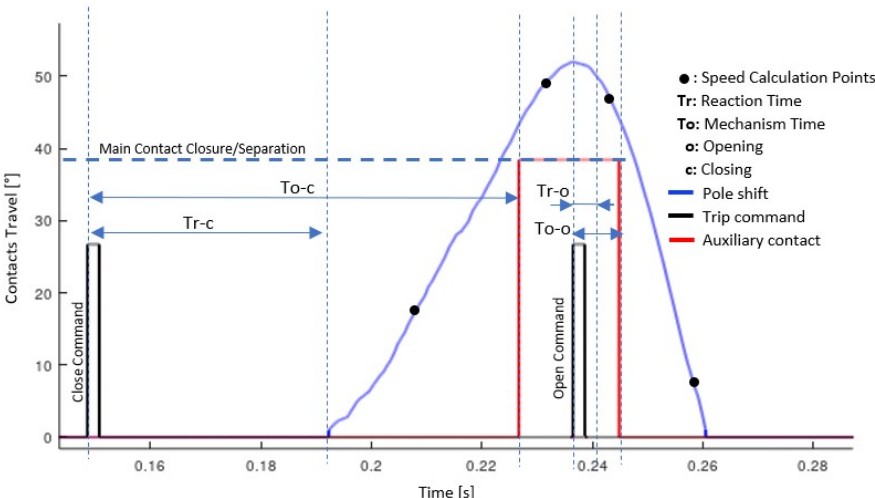

**Figure 1.** Contact displacement curve for an O-C-O opening–closing maneuver.

## 3. Laboratory Testing System

For the laboratory tests, the signals were obtained through accelerometers and incremental encoders installed in a three-phase circuit breaker GL312 from Alstom—GE Grid Solution, for voltages of 145 KV, with a nominal current of 3150 A and a breaking capacity of 40 KA. Its data are presented in Table 1 and its installation in the laboratory in Figure 2. The incremental encoders are from Sick, model DFS60B-S4PA10000, with a maximum resolution of 10,000 pulses per revolution and were installed on the axes of each pole, as shown in Figure 3. The accelerometers, brand PHD, model SI110A, with a sensitivity of 100 mV/g, maximum amplitude of 80 g and frequency response from 0.7 Hz to 15,000 Hz, were installed at the base of each pole, in the horizontal direction, in order to eliminate their saturation, as illustrated in Figure 4.

**Table 1.** Technical Characteristics of the HVCB GL 312.

| GL 312 Technical Characteristics | |
|---|---|
| Rated voltage | 145 kv |
| Rated frequency | 50/60 Hz |
| Rated normal current | up to 3150 A |
| Rated short-circuit breaking current | up to 40 kA |
| Rated short-circuit making current | 104 kA |
| Rated duration of short-circuit | 3s |
| Opening time | 28 ms |
| Break time | 50 ms |
| Closing time | ≤70 ms |

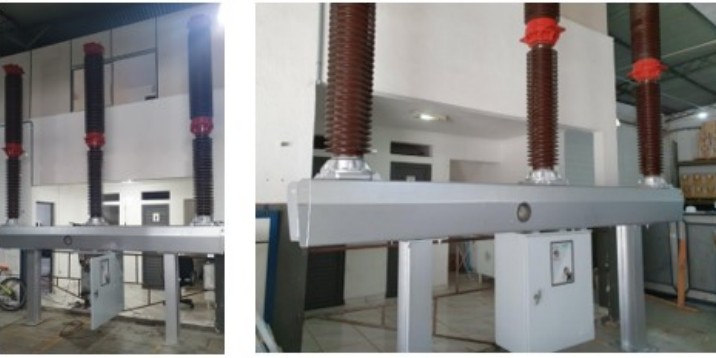

**Figure 2.** CGL 312 test circuit breaker.

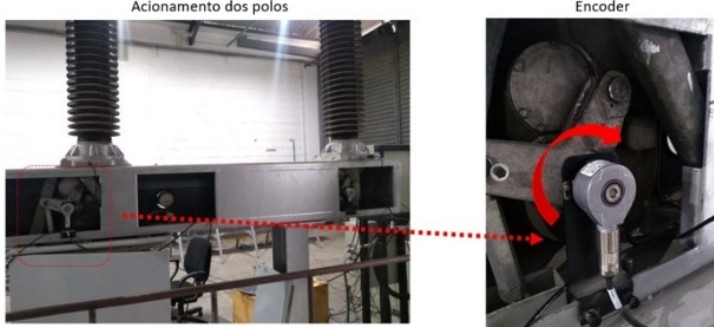

**Figure 3.** Incremental encoders.

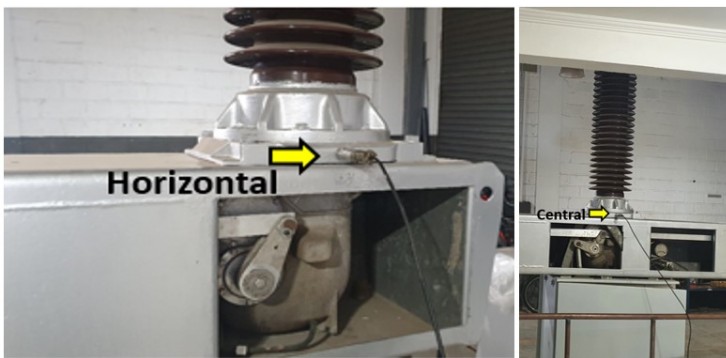

**Figure 4.** Installed accelerometers.

The signals were obtained via the Preditor 4.0® acquisition system, which has an analog–digital converter with 24-bit resolution and a sampling frequency of 46.875 kHz, as shown in Figure 5. The acquisition system in question was configured to start the acquisition automatically; that is, when any channel receives a signal that exceeds the preset threshold (defined by the user) the signals of all channels will be recorded, in an acquisition window of 10 s. Furthermore, there is a routine so that the acquisitions are not overwritten.

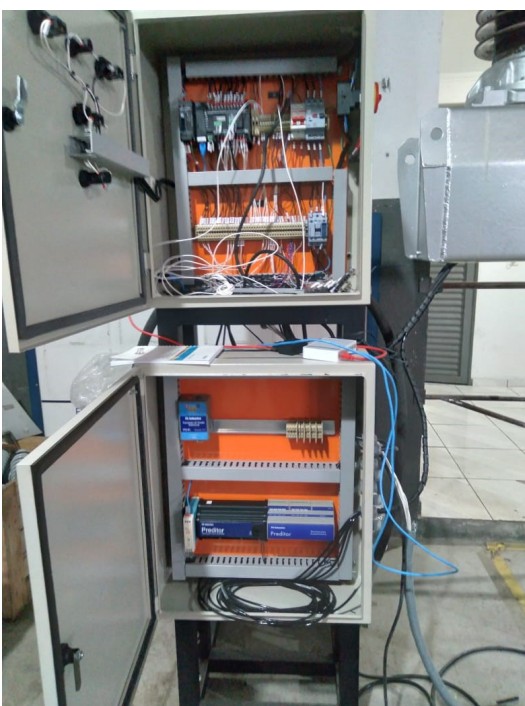

**Figure 5.** Preditor 4.0 acquisition system.

## 4. Result

*4.1. Mechanism Time Detection via Vibration Signal and Auxiliary Contact*

With the circuit breaker acceleration signals and auxiliary contact signals, the breaker mechanism times will be obtained and compared with the contact displacement curve. It should be added that, in this study, a circuit breaker with only one auxiliary contact was used; thus, the mechanism time for each pole will not be calculated, but, rather, the mechanism time with the joint activation of all poles. As shown in Figure 1, the mechanism time for a closing operation is determined by the difference between the instant of the close command and the instant of contact closure. In this way, with the signal obtained via the auxiliary contact, there is the moment of closing the contact. Therefore, in order to determine mechanism time, it will be necessary to determine, with the acceleration signals, only the moment of the closing command (close command).

In Figure 6, the collected signals of an opening operation are presented with emphasis on the moment of the closing command (close command). The acceleration and displacement signals are observed for each pole, as well as the trip signal and the auxiliary contact signal.

It can be seen in Figure 6 that the vibration signals of the two accelerometers responded to the instant of the closing command, due to the mechanical actuation of the trip relay. In Figure 7, the same signals are shown close to the moment of the closing command, where the occurrence of a transient at the moment of the command is observed, however, with low amplitude, close to noise. It is also observed in this figure that the noise is common for both accelerometers, since they come from the network, while the transient associated with the relay actuation presented different levels between the accelerometer of one pole and the other. The reason for this discrepancy is the location of the accelerometers relative to the location of the breaker trip relay. As the accelerometers are installed on the poles and the trip relay is on the circuit breaker control panel (Figure 2), it can be seen that the accelerometer located on the central pole (pole B) will be more sensitive to the actuation of the trip relay than the other accelerometers. For this reason, the transient accelerometer signal from pole B (in red in the graphs) has greater amplitude than the accelerometer signal from pole A (in blue in the graphs).

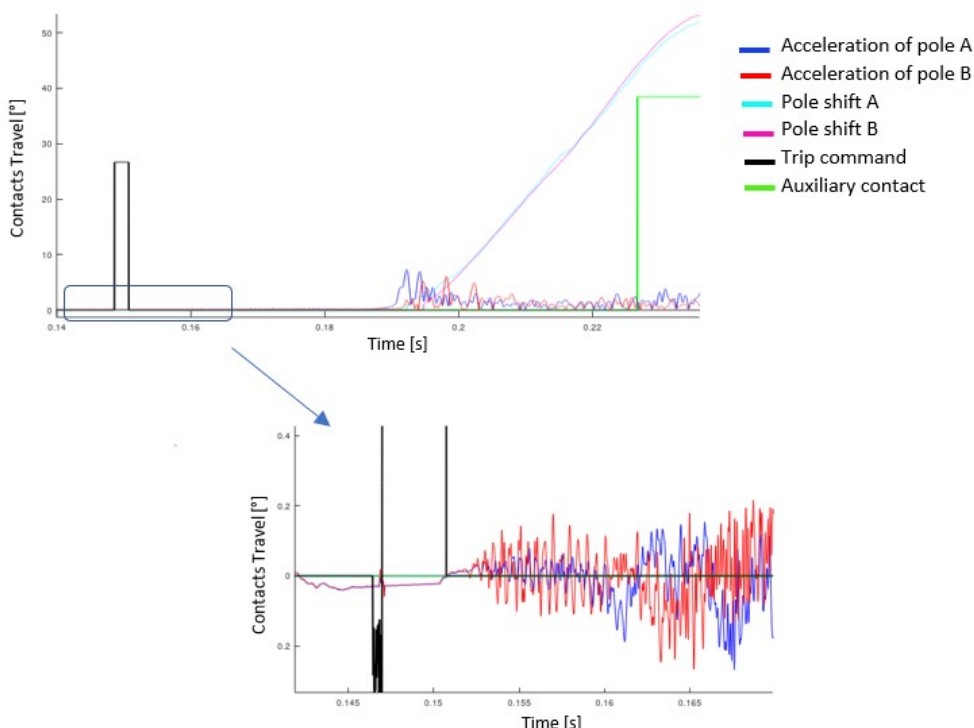

**Figure 6.** Instantaneous closing command (part 1).

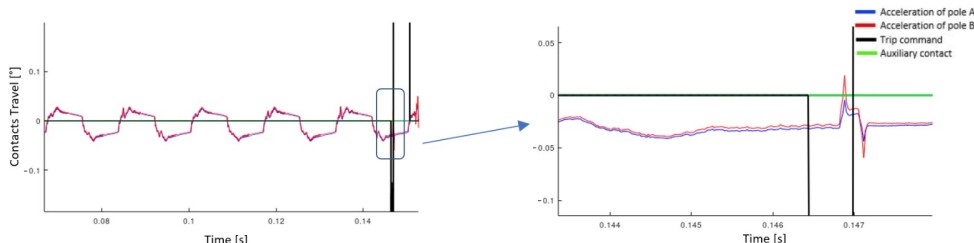

**Figure 7.** Instantaneous closing command (part 2).

From the observations presented, it appears that the difference between the acceleration signals will be useful in defining the instant of the closing command. Initially, to determine the instant, the velocity signal was obtained for each accelerometer, with the integral of each signal. Subsequently, the difference between the speed signals was performed, which resulted in the signal in Figure 8. In the signal presented, it is observed that the first positive peak occurs at the instant of the circuit breaker command (0.147 s).

With the instant of the closing command (0.147 s) defined and the instant of closing of the contacts (0.226 s) obtained via the auxiliary contact of the circuit breaker (Figure 8), it is possible to calculate the time of the mechanism of the circuit breaker, that being the difference between these instants: 0.079 s.

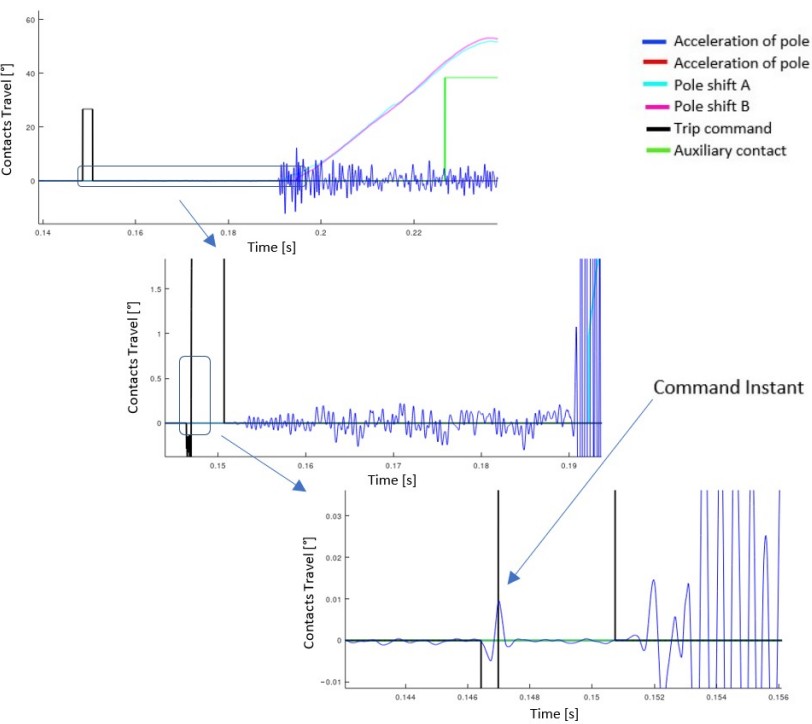

**Figure 8.** Instant of the closing command, highlighting the difference between the acceleration signals of poles A and B.

### 4.2. Obtaining Reaction Time via Vibration Signal

In this section, the method for obtaining the reaction time for opening and closing using vibration signals will be presented. The signals collected from an opening operation are shown in Figure 9, highlighting the opening moment. The acceleration and displacement signals are observed for each pole, as well as the trip signal and the auxiliary contact signal.

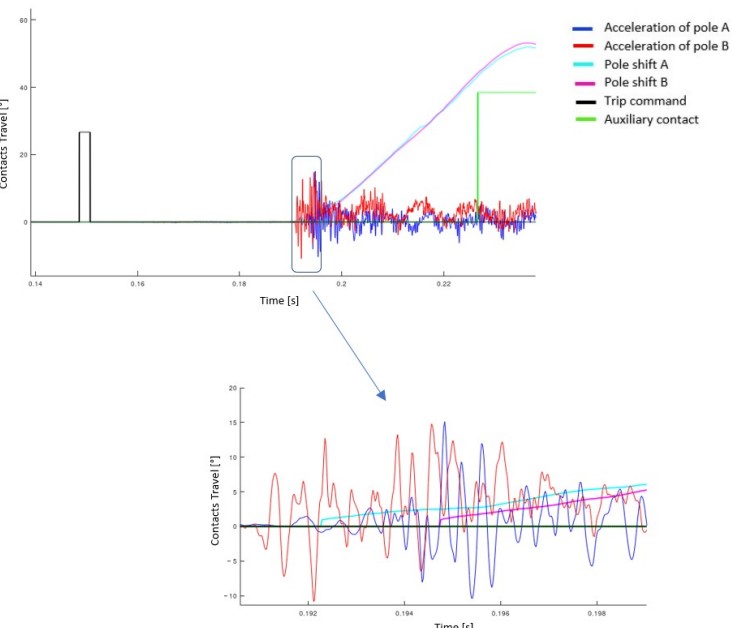

**Figure 9.** Opening maneuver.

It can be seen in Figure 9 that the vibration level becomes high when the movement of the contacts starts; however, it is not possible to define with these vibration signals the

precise moment of the beginning of the movement. For the treatment of the vibration signal, the envelope technique will be used, which consists of applying a band-pass filter, tuned in the resonance band of the signal at the moment of the maneuver. The resulting signal is demodulated amplitude using the Hilbert transform modulus.

Figure 10 shows the spectrum of the acceleration signal at the moment of the maneuver, where the concentration of components in the band from 500 to 2500 Hz was identified. Therefore, this frequency range will be used as the resonance band for the calculation of the envelope.

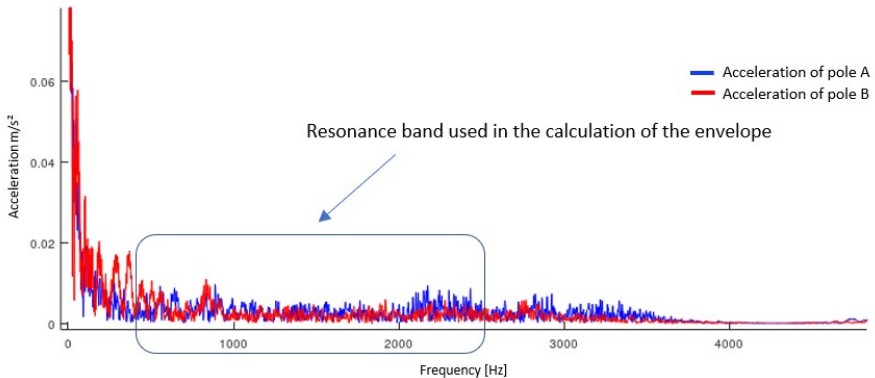

**Figure 10.** Spectrum of acceleration signals at the moment of the maneuver, highlighting the resonance band.

Figure 11 below shows the previous vibration signals after applying the envelope technique. Based on the envelopes obtained, the first peak of the envelopes shows the instant when the contacts began to move, for the two poles of the circuit breaker, those being 0.1922 s for pole A and 0.1942 for pole B. Considering the close command time of 0.147 s, obtained in the previous section, the closing reaction time for pole A is 0.0452 s and for pole B is 0.0472 s.

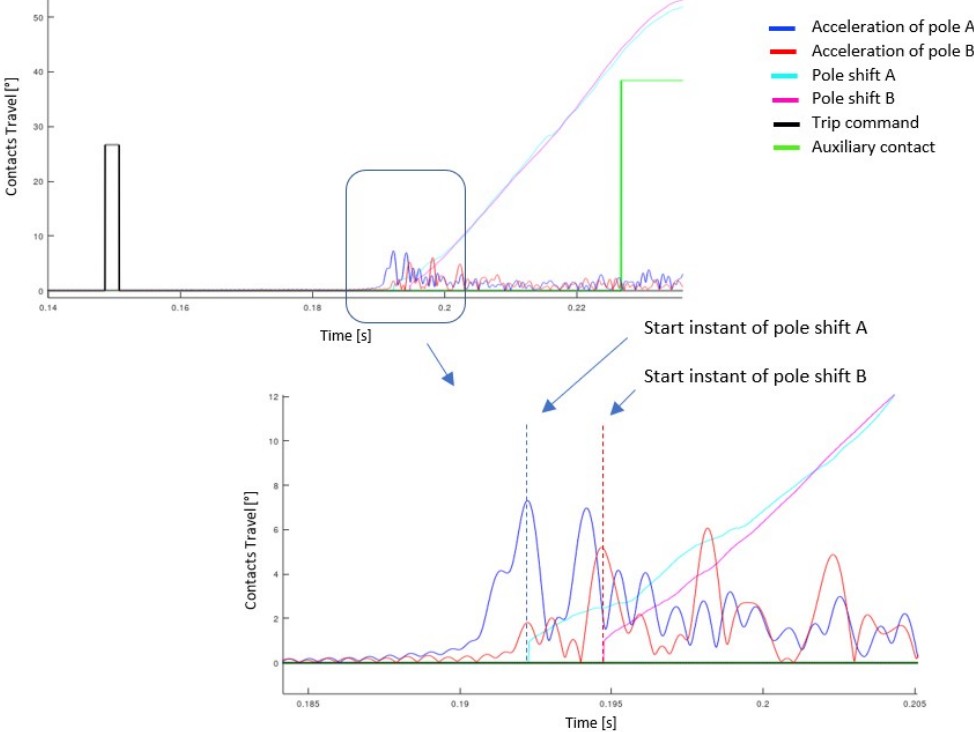

**Figure 11.** Opening with acceleration envelopes.

As shown in the results, for each pole, the respective reaction time of a closing operation was obtained, using only vibration signals. The reaction time was obtained via the difference between two instants identified in the vibration signals: the trip instant and the contact movement initiation instant (for each pole).

The trip instant (or command instant) was obtained via the difference between the acceleration signals of each pole, where the first peak of this resulting signal marks the respective instant. This was based on the fact that one of the accelerometers (central pole) is located in the part of the circuit breaker structure that is closest to the trip relay; therefore, the accelerometer associated with it was more sensitive to this event. In addition, the difference between the acceleration signals of the poles makes it possible to attenuate noises that are present in both. For the demonstration of the trip instant, the trip signal obtained superimposed on the acceleration signals and the signal resulting from the subtraction of the respective acceleration signals was presented (Figures 6–8).

In order to elucidate the techniques developed in this work to obtain the HVCB mechanism and reaction time, Figure 12 shows a flowchart with the mentioned stages of processing the acceleration signals.

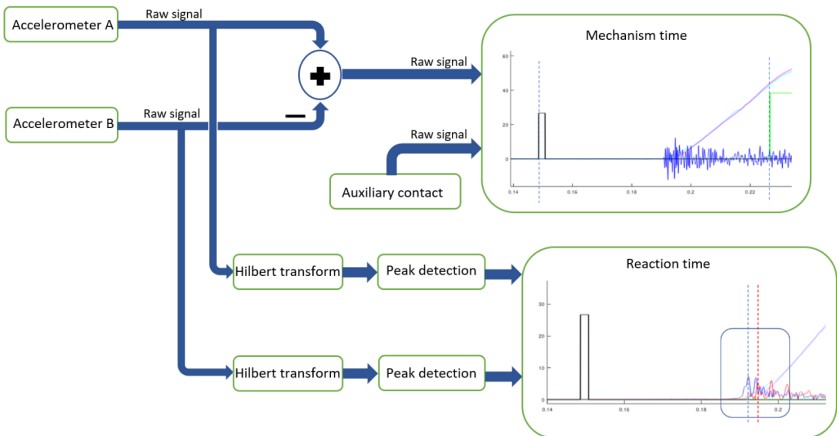

**Figure 12.** Flowchart with the processing steps of the acceleration signals to obtain the HVCB mechanism and reaction time.

The instant of contact movement was obtained only via acceleration signals, without the need for encoders installed in the moving parts of the circuit breaker. This instant was calculated by applying the envelope technique to the acceleration signal of each pole, with the first peak of this resulting signal being the instant of contact movement. The demonstration of this instant is presented via signals obtained from encoders installed in the circuit breaker, superimposed on the accelerometer signals and the signals obtained via the envelope technique (Figures 9 and 11).

## 5. Field Installation of Monitoring System

### 5.1. Structure and Functioning of the Monitoring System

This chapter presents the monitoring system of an HVCB implemented in the field. The installation of the system is in an HVCB, isolated to SF6, model GL 314X, manufactured by Alstom. Its data are presented in Table 2. This HVCB, illustrated in Figure 13, operates in the 525/230 kV substation of Bateias-PR, which is managed by Copel Geração e Transmissão S.A. and is responsible for inserting a bank of capacitors, as shown in Figure 14.

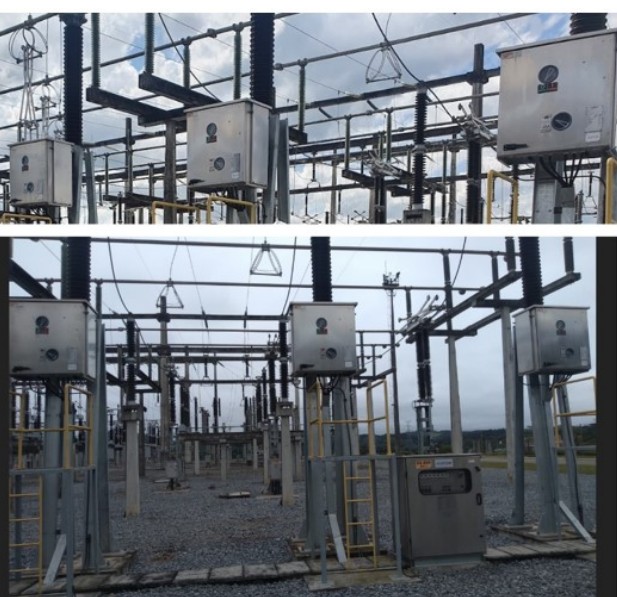

**Figure 13.** GL 314X circuit breaker, 525/230 kV substation, Copel Geração e Transmissão S.A (Copel Generation and Transmission), Bateias-PR.

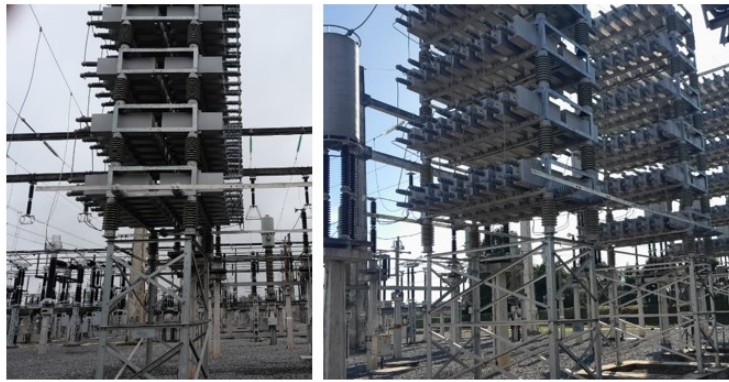

**Figure 14.** Bank of capacitors connected to the HVCB GL 314X.

**Table 2.** HVCB GL 314X data.

| | Description | GL314X |
|---|---|---|
| | Norma | CEI |
| Ur | Nominal voltage (effective value) | 245 KV |
| Ir | Rated current in continuous service | 4000 kV |
| fr | Nominal frequenc | 60 Hz |
| Ip | Crest value of nominal withstand current | 125 kA |
| Ik | Rated short-time withstand current | 50 kA |
| Ud | Rated lightning impulse withstand voltage (crest value) | 460 kV |
| Up | Phase to earth and between phases through open contacts | 1050 kV |
| Isc | Rated symmetric interruption current | 50 kA |
| | Rated establishment current (crest value) | 110 kA |
| | Total interruption time | $50 \pm 2$ ms |
| | Nominal sequence of operation | O—O.3 s—CO—3 min—CO |
| | Resistance of main contacts (new contacts) | $\leq 44$ v$\Omega$ |

The online monitoring system for the operational condition of the implemented HVCB seeks the early identification of problems that may affect its interruption capacity. For this, instrumentation was implemented that is capable of collecting signals that are associated with mechanical and/or electrical wear from gaps in its mechanical structure and signals that are associated with loss of insulating material (such as leakage of SF6 gas). In the

developed monitoring system, two breaker operation events were considered that will generate the signals used in the breaker evaluation, the switching event and the loading event. Thus, for an operation event, an acquisition will be triggered to collect monitoring signals from the poles: vibration (accelerometers), contact displacement (linear transducer), opening and closing contact (auxiliary contact), tripping current (transducer current) and extinguishing current (current transformer). In cases of a charging event, the system will collect the signals associated with the circuit breaker charging system: vibration (accelerometers), charging motor current (Hall effect current transformer) and charging motor supply voltage (transformer Hall effect voltage).

For the presentation of the sensors used in the monitoring, their locations are shown in Figure 15.

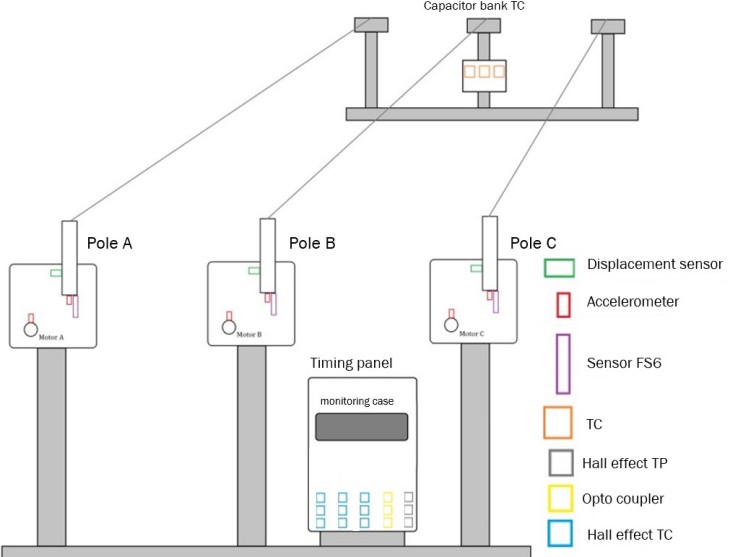

**Figure 15.** Locations of the monitoring system sensors.

As previously mentioned, there is the operation event, of short duration, less than 1 s, in relation to the monitored events, which demands a precise and sensitive follow-up to trigger the acquisition of the signals, in addition to a sufficient sampling frequency, in order to obtain a sufficient amount of samples for the construction of the signals generated by each of the sensors. The switching events consist of the mechanical displacement of the circuit breaker contacts. Thus, as shown in Figure 16, the signals arising from this event are:

- Opening trip signal: Signal resulting from an opening maneuver; in this scenario, the circuit breaker synchronism panel receives a command, a voltage level, which is applied to the contactor responsible for opening. The trip signal is then obtained via a Hall-type current transformer placed in the respective conductor, represented by the consequent current of this command.
- Closing trip signal: Signal resulting from a closing maneuver; in this scenario, the synchronism panel receives a command, a voltage level, which is applied to the contactor responsible for closing. The trip signal is then obtained via a Hall-type current transformer placed in the respective conductor, represented by the consequent current of this command.
- Contact displacement signal: Signal resulting from an opening or closing maneuver, when the main shaft of the mechanism rotates, effecting the displacement of the contacts. This movement is obtained via a linear displacement sensor, installed on the breaker shaft.
- Pole acceleration signal: Signal resulting from an opening or closing maneuver, described in the previous topics. The vibration caused by this movement is obtained via an accelerometer installed in the lower region of the circuit breaker housing.

- Extinguishing current signal: Current signal obtained at the instant of opening or closing of the breaker. This current is obtained via sensors installed on the secondary of the CT located on the circuit breaker phases.
- Auxiliary contact signal: Signal corresponding to the status of the breaker contact, open or closed. This is obtained through an optocoupler connected to the auxiliary contact of the circuit breaker. The breaker auxiliary contact terminals are available inside the breaker's synchronism panel.
- SF6 sensor signal: Signal obtained via a multiparameter sensor installed in the circuit breaker housing. This signal provides information regarding the level of contamination of the SF6 gas.

The second monitored event is the loading event, which lasts longer than the operation event; it exceeds 10 s; thus, in addition to a sufficient sampling frequency, storage with a memory level adequate to the duration of this event is required. The loading events, as illustrated in Figure 16, consist of the loading of the circuit breaker spring by a DC-powered motor; thus, the resulting signals are as follows:

- Motor voltage signal: Voltage signal applied to the motor during the charging event. This signal is obtained via a Hall-type voltage sensor and the sensor connection points are inside the synchronism panel.
- Motor current signal: Current signal from motor energization during charging event. This signal is obtained via a Hall-type current sensor and the conductors for connecting the sensors are inside the synchronism panel.
- Motor acceleration signal: Vibration signal resulting from motor running during the loading event. This signal is obtained via an accelerometer that is installed in the motor frame, in the pole panel.

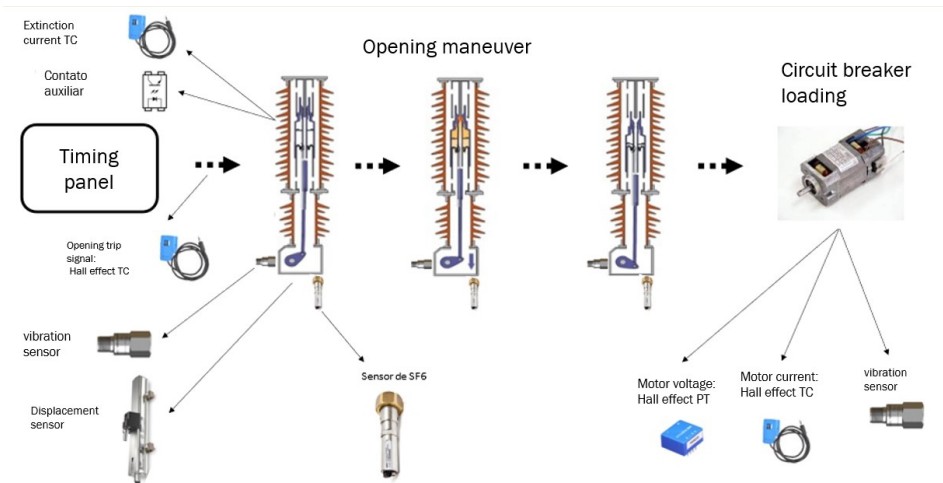

**Figure 16.** Signals obtained during opening and loading maneuvers.

Note the simplicity of the proposed tool, as it is based on the temporal analysis of signals without the need for transformations or advanced digital signal processing techniques. Furthermore, this makes the analysis straightforward, reducing the need for signal feature interpretation and can be easily automated by an algorithm. Finally, the implementation of the subtraction between the acceleration signals results in a significant reduction in noise shared between the accelerometers.

*5.2. Monitoring System Hardware and Sensors*

In the monitoring system, the occurrence of opening, closing or loading events, the signals generated by the sensors will be collected and stored by the acquisition hardware, the Preditor 4.0® system, illustrated in Figure 17. This acquisition system consists of 32 channel analog readouts with 24-bit resolution and a sampling frequency of 46.875 kHz.

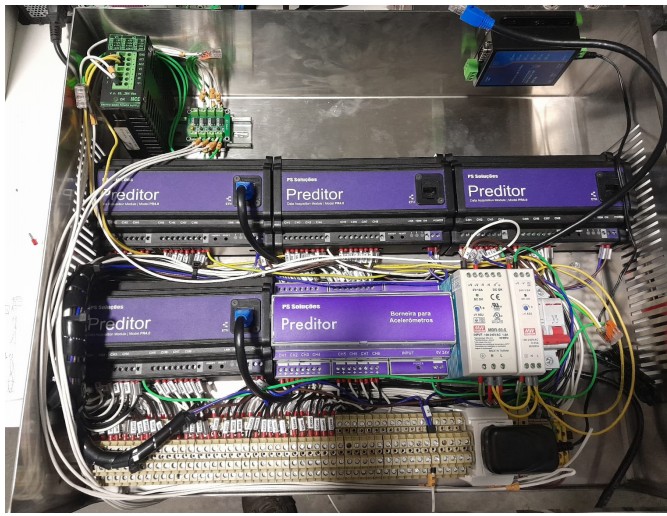

**Figure 17.** Preditor 4.0 data acquisition system.

For monitoring the HVCB, the level of invasiveness of the installation was minimized, using window-type current sensors and installation of linear encoders at strategic points, which do not require the disassembly of circuit breaker components.

When monitoring the poles, the installation of a magnetostrictive linear encoder, model F305785 manufactured by Gefran®, was considered, and used for commissioning tests of circuit breakers (Figure 18). Figure 18 shows the fixing of the encoder to the shaft of the crankcase by a 7075 aluminum rod, as well as fixing the encoder to the pole structure with a guide made of steel.

The installation point of the accelerometer was situated in the lower region of the crankcase. The installation of the multiparameter sensor/transmitter, DPT145 for Vaisala® SF6 gas, permitted the online measurement of dew point, pressure and temperature. Figure 19 illustrates the pole crankcase with the respective installed sensors.

In monitoring the spring loading of the circuit breaker, the installation of an accelerometer in the frame of the spring loading motor was defined. In order to obtain the extinction current of the circuit breaker, an analysis was made of the location for accommodating the current sensor, responsible for collecting the secondary current of the CT connected to the HVCB. The breaker command signals (trip on and trip off) and current signals from the HVCB charging motor were collected via Hall-type clamp on current transducers, installed in conductors that are allocated in the HVCB synchronism panel, as shown in Figure 20.

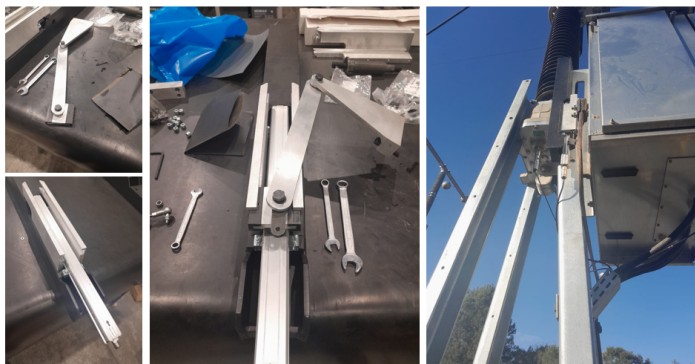

**Figure 18.** Linear encoder installed on the pole.

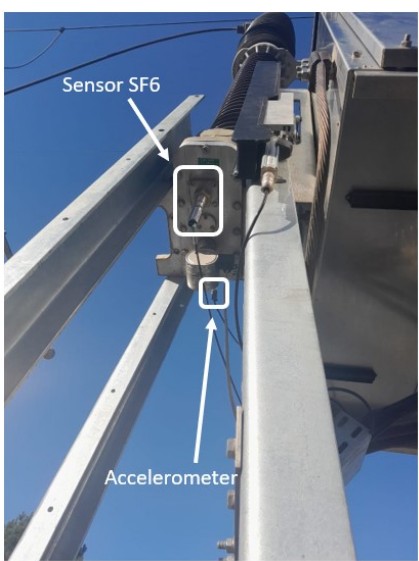

**Figure 19.** SF6 sensor and accelerometer installed in the pole crankcase.

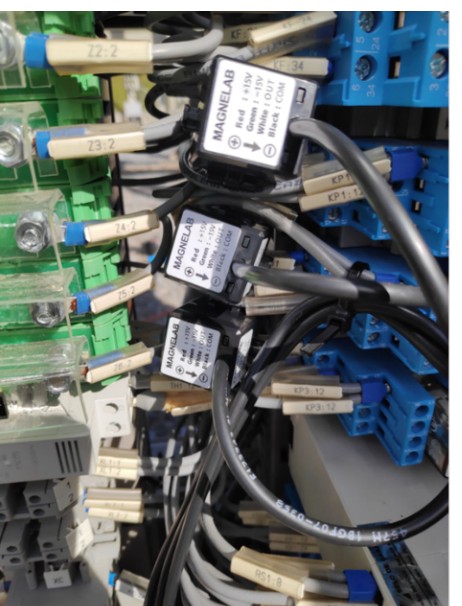
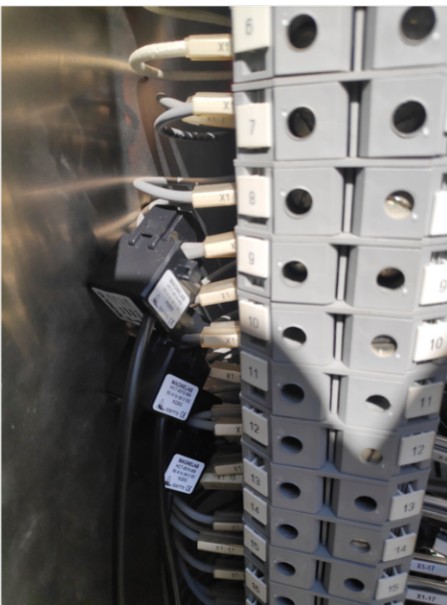

**Figure 20.** Hall-type clamp on current transducers installed in the HVCB synchronism panel.

It was observed by Copel's technical team that the breaker's synchronism panel had enough space to accommodate the monitoring system case, as shown in Figure 21. Thus, analyses were carried out on the passage of instrumentation cables between the panels from each pole to the synchronism panel, in addition to passing the signal cable from the CT to the synchronism panel. With this analysis, the option of passing cables in the same cable channel as the circuit breaker was noted, discarding the need for additional infrastructure for this purpose.

As shown, the implementation of the online monitoring system did not require structural changes to the HVCB, such as the installation of external panels and the passage of cables through additional ducts. Furthermore, the installed sensors did not require the de-energization of the synchronism panel nor the terminal disconnection. The monitoring system made it possible to detect any mechanical and/or electrical wear on the HVCB through the following diagnoses:

- Characterization of degradation in contact resistance of HVCB poles via online monitoring of their wear.

- Characterization of HVCB insulator loss degradation characterization via online monitoring of opening and closing times and SF6 gas condition.
- Characterization of the degradation of the mechanical part of the SF6 circuit breaker via online monitoring of the actuation current via electrical signature.

In this section, it is important to highlight the ease of implementing instrumentation in circuit breakers with the proposed tool. This is because, with traditional monitoring methods, it would be necessary to install encoders directly on the pole axis to obtain the instants of movement initiation. Furthermore, as in the tool proposed in the previous section, the signals are analyzed in the time domain; thus, an algorithm for automating this process is simple; its main function would be the detection of the first peak of the signal from each accelerometer, after the presented demodulation process.

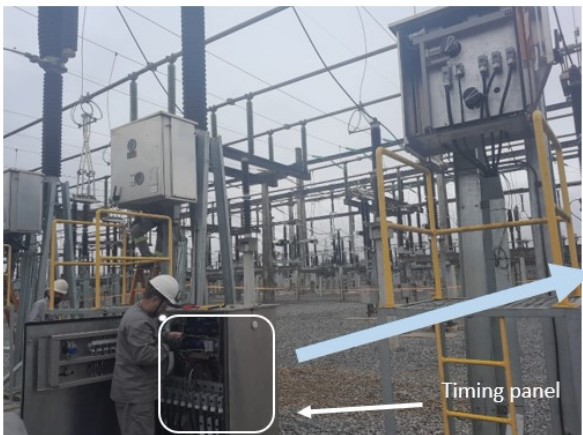 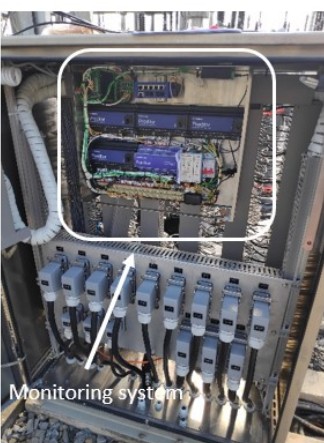

**Figure 21.** Case of the monitoring system installed in the HVCB synchronism panel.

## 6. Conclusions

Condition-based maintenance on outdoor high voltage SF6 circuit breakers (HVCBs) is an important issue for the future of Smart Grids. For the evaluation of the condition of an HVCB, mechanism, insulation and contacts, the measurement of time and kinetic parameters by means of the travel curve is a competent method. However, access to the main contacts of each circuit breaker pole is inapplicable for online evaluation. Thus, this work presented a solution to obtain the parameters necessary for online monitoring in a non-invasive way. The method is based on obtaining the mechanism time and the reaction time of each pole of the HVCB using only acceleration and auxiliary contact signals. Experiments were conducted in an HVCB test laboratory with accelerometers and angular encoders and the results obtained with the vibration analysis, compared with those obtained via angular encoders, pointed to the effectiveness of the method in obtaining temporal performance parameters of an HVCB from its travel curve (mechanism and reaction time). In this work, the signals are processed in the time domain, which makes its automation simple, since it only depends on a function to detect the first peak of a signal obtained via the accelerometer. Compared to traditional methods of monitoring outdoor SF6 circuit breakers that are based on encoders, the proposed method presented ease of installation, as it only requires the insertion of accelerometers. The implementation of the field monitoring system was presented, containing additional instrumentation for the acquisition of trip current signals, the SF6 gas condition and extinguishing current signals. The system implemented in the field did not require structural changes to the HVCB, such as the installation of external panels and the passage of cables through additional ducts. Furthermore, the installed sensors did not require the synchronism panel to be de-energized or the terminals to be disconnected. Thus, it was possible to detect possible mechanical and/or electrical wear on the outdoor SF6 circuit breaker associated with degradation in the contact resistance of the poles, loss of the insulating characteristic of the SF6 gas and degradation of the mechanism.

**Author Contributions:** Conceptualization, C.L.D.S., F.d.O.A. and G.L.-T.; methodology, O.R., F.d.O.A. and G.L.-T.; software, I.A.d.-S.A. and E.L.B.; validation, E.L.B., L.E.B.-d.-S. and L.E.L.O.; formal analysis, C.L.D.S., F.d.O.A. and G.L.-T.; investigation, C.L.D.S., F.d.O.A., J.C.O.C., R.M. and G.L.-T.; resources, C.L.D.S., J.C.O.C. and R.M.; data curation, C.E.X., E.L.B., L.E.B.-d.-S. and L.E.L.O.; writing—original draft preparation, C.L.D.S., F.d.O.A. and G.L.-T.; writing—review and editing, C.L.D.S., F.d.O.A. and G.L.-T.; visualization, C.E.X., E.L.B., L.E.B.-d.-S. and L.E.L.O.; supervision, C.L.D.S. and R.M.; project administration, C.L.D.S. and G.L.-T.; funding acquisition, C.L.D.S. All authors have read and agreed to the published version of the manuscript.

**Funding:** This paper presents part of the results obtained during the execution of the R&D project of code PD-06491-0511/2018 titled "MONITORING SYSTEM AND EVALUATION OF THE REDUCTION OF THE INTERRUPTION CAPACITY OF CIRCUIT BREAKERS", executed by the Gnarus Institute for Copel Geração e Transmissão S.A., the project's financier, under the Research and Development Program of the Brazilian Electric Sector, regulated by the National Electric Energy Agency—ANEEL.

**Institutional Review Board Statement:** Not applicable.

**Informed Consent Statement:** Not applicable.

**Data Availability Statement:** Not applicable.

**Conflicts of Interest:** The authors declare no conflict of interest.

## Abbreviations

The following abbreviations are used in this manuscript:

| | |
|---|---|
| HVCB | High Voltage Circuit Breaker |
| SF6 | Sulfur hexafluoride |
| CIGRE | Conseil International des Grands Resaux Electriquies |
| FFT | Fast Fourier Transform |
| ZFFT | Zoom FFT |
| CZT | Shirp z transform |
| STFT | Short Time Fourier Transform |
| EMD | Empirical Mode Decomposition |
| CWT | Continuous Wavelet transform |
| OC | Opening–closing |
| OCO | Opening–closing-opening |
| Tr-c | Closing reaction time |
| Tr-o | Opening reaction time |
| To-c | Mechanism closing time |
| To-o | Mechanism opening time |
| CT | Current transformer |
| DC | Direct current |
| PT | Potential transformer |

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
