# Peer review of "An Online Non-Invasive Condition Assessment Method of Outdoor High-Voltage SF6 Circuit Breaker"

_machines, doi:10.3390/machines11030323_

Round 1

Reviewer 1 Report

The mechanical reliability of the in-service high voltage circuit breaker draws a wield interest all over time. It is significant to detect a potential major or minor failure of the HVCB by a series of non-invasive measures. Engineers dedicated to electrical engineering and switching equipment could have a vast interest in the work of the manuscript. I have some questions and suggestions that need to be discussed as follows:

1.     The novelty of the study needs to be clarified, by comparing past online monitoring technologies with the proposed online non-invasive condition assessment method. From my point of view, there are lots of online invasive monitoring devices that are installed in GIS (Metal Enclosed and Gas Insulated Switchgear) and be in service for several years, such as vibration-detecting devices, travel curve measuring systems, noise-detecting devices, temperature rising detecting sensors and system, et al., and some of their integrates.

2.     The manuscript in its current version looks like a technic report more than a scientific paper. The authors should be more focused on presenting the innovative online invasive assessment method or illustrating the key technologies of invasive assessment devices.

3.     Some of the terminologies should be constant and suggested to follow the current IEC or IEEE international standards for an easy understanding of the phrases. Such as the following terminologies (listed but not limited in this comments sheet):

1)     “mobile contacts” on page 2 line 41 should be “movable contacts” or “moving contacts”

2)     “ opening maneuver” on page 3 line 89 should be “opening operation”, and so on.

Author Response

Reviewer #1

The authors thank to honorable reviewer for excellent review of the paper and nice suggestions made to improve the quality of the paper. Followings are the response to the suggestions of honorable reviewer. The valuable suggestions and corrections are incorporated cautiously, and the paper is made clearer while revising the manuscript.

Question 1

For compliance with this item, the work was restricted to Outdoor High-Voltage SF6 Circuit Breaker. Highlighting that the work involves Outdoor High-Voltage SF6 Circuit Breaker.

Question 2

From this item, the work was updated, where the following paragraphs were added in the sections: 4.1. Mechanism time detection by vibration signal and auxiliary contact, and 4.2. Obtaining reaction time by vibration signal:

Note the simplicity of the proposed tool, as it is based on the temporal analysis of signals without the need for transformations or advanced digital signal processing techniques. Furthermore, this makes the analysis straightforward, reducing the need for signal feature interpretation and can be easily automated by an algorithm. Finally, the implementation of the subtraction between the acceleration signals results in a significant reduction of noise shared between the accelerometers.

In this section, it is important to highlight the ease of implementing instrumentation in circuit breakers with the proposed tool. Because, with traditional monitoring methods, it would be necessary to install encoders directly on the pole axis to obtain the instants of movement initiation. Furthermore, as in the tool proposed in the previous section, the signals are analyzed in the time domain, thus, an algorithm for automating this process is simple, whose main function would be the detection of the first peak of the signal from each accelerometer, after the presented demodulation process.

Question 3

The expressions "mobile contacts" and "opening maneuver" were updated to "moving contacts" and "opening operation", respectively. The following table has been added to suit this item:

Abbreviations

HVCB  High Voltage Circuit Breaker

SF6     Sulfur hexafluoride

CIGRE Conseil International des Grands Resaux Electriquies

FFT     Fast Fourier Transform

ZFFT   Zoom FFT

CZT     Shirp z transform

STFT   Short Time Fourier Transform

EMD    Empirical Mode Decomposition

CWT    Continuous Wavelet transform

OC      Opening-closing

OCO    Opening-closing-opening

Tr-c      Closing reaction time

Tr-o     Opening reaction time

To-c     Mechanism closing time

To-o     Mechanism opening time

CT       Current transformer

DC       Direct current

PT       Potential transformer

Reviewer 2 Report

The authors present a method based on obtaining the mechanism time and the reaction time of each pole of the HVCB using only acceleration and auxiliary contact signals. Experiments were conducted in an HVCB test laboratory, instrumented with accelerometers and angular encoders, and the results obtained with the vibration analysis, compared with those obtained by angular encoders, pointed to the effectiveness of the method in obtaining temporal performance parameters of a HVCB from its travel curve (mechanism and reaction time).

Some comments next:

1)      The references section is poor. There are 14 references of the 23 with more than ten years old. Maybe, some of these could be updated.

2)      Importantly novelty aspect is not very much evident, and a convincing statement or results could not be seen regarding this.

3)      How was chosen the two accelerometers were installed on the base of two poles of the circuit breaker?

4)      The experimental section has not been described adequately.

Are the variances of the measurements adequate for the fault diagnosis? What is the influence of the noise? Which are the warning/alarm levels of this indicator? There are too many unanswered questions that must be fully addressed to validate a new diagnostic method, which is not even mentioned in the presented work

Author Response

Authors are happy to note that the reviewer appreciates and understands the current work. Point by point response to various comments is given below and proposed changes have been made in the revised manuscript.

Question 1

The old references have been updated, with the new references presented below:

[8] R. F. Ribeiro Junior, I. A. dos S. Areias, M. M. Campos, C. E. Teixeira, L.E. Borges da Silva, G. F. Gomes. Fault detection and diagnosis in electric motors using convolution neural network and short-time fourier transform. Journal of Vibration Engineering & Technologies, 2022.

[9] Areias, I.A.d.S.; Borges da Silva, L.E.; Bonaldi, E.L.; de Lacerda de Oliveira, L.E.; Lambert-Torres, G.; Bernardes, V.A. Evaluation of Current Signature in Bearing Defects by Envelope Analysis of the Vibration in Induction Motors. Energies 2019, 12, 4029.

 [7]: A. Asghar Razi-Kazemi and K. Niayesh, "Condition Monitoring of High Voltage Circuit Breakers: Past to Future," in IEEE Transactions on Power Delivery, vol. 36, no. 2, pp. 740-750, April 2021, doi: 10.1109/TPWRD.2020.2991234.

 [10]: J. Yang, G. Zhang, B. Chen and Y. Wang, "Vibration signal augmentation method for fault diagnosis of low-voltage circuit breaker based on W-CGAN," in IEEE Transactions on Instrumentation and Measurement, doi: 10.1109/TIM.2023.3240228.

 [12]: H. Balan, T. Varodi, M. I. Buzdugan and R. A. Munteanu, "Monitoring power breakers using vibro acoustic techniques," 2016 International Conference on Applied and Theoretical Electricity (ICATE), Craiova, Romania, 2016, pp. 1-6, doi: 10.1109/ICATE.2016.7754664.

 [14]: K. Obarcanin, "A High Voltage Circuit Breaker Behavior Change Indices based on the Vibration Signature Analysis," 2020 19th International Symposium INFOTEH-JAHORINA (INFOTEH), East Sarajevo, Bosnia and Herzegovina, 2020, pp. 1-5, doi: 10.1109/INFOTEH48170.2020.9066279.

 [15]: Q. Yang, J. Ruan, Z. Zhuang and D. Huang, "Condition Evaluation for Opening Damper of Spring Operated High-Voltage Circuit Breaker Using Vibration Time-Frequency Image," in IEEE Sensors Journal, vol. 19, no. 18, pp. 8116-8126, 15 Sept.15, 2019, doi: 10.1109/JSEN.2019.2918335.

 [17]: A. A. Razi-Kazemi, M. Vakilian, K. Niayesh and M. Lehtonen, "Circuit-Breaker Automated Failure Tracking Based on Coil Current Signature," in IEEE Transactions on Power Delivery, vol. 29, no. 1, pp. 283-290, Feb. 2014, doi: 10.1109/TPWRD.2013.2276630.

 [18]: H. Ahmad and T. S. Kiong, "Trip Coil Signature Measurement and Analysis Techniques for Circuit Breaker," 2016 7th International Conference on Intelligent Systems, Modelling and Simulation (ISMS), Bangkok, Thailand, 2016, pp. 261-267, doi: 10.1109/ISMS.2016.28.

[19] Jianzhong Zhang, Yongbin Wu, Zheng Xu, Zakiud Din, Hao Chen,” Fault diagnosis of high voltage circuit breaker based on multi-sensor information fusion with training weights”, Measurement,Volume 192,2022.

 [21] Q. Yang, J. Ruan, Z. Zhuang and D. Huang, "Fault Identification for Circuit Breakers Based on Vibration Measurements," in IEEE Transactions on Instrumentation and Measurement, vol. 69, no. 7, pp. 4154-4164, July 2020.

 [23] Q. Yang, J. Ruan, Z. Zhuang and D. Huang, "Chaotic Analysis and Feature Extraction of Vibration Signals From Power Circuit Breakers," in IEEE Transactions on Power Delivery, vol. 35, no. 3, pp. 1124-1135, June 2020.

Question 2

To comply with this item, the following changes were made:

Abstract: Online monitoring of outdoor high voltage SF6 circuit breakers (HVCB) is essential to detecting potential damages. To this end, the study of accurate and non-invasive monitoring methods has been significantly investigated in recent decades. Considering that HVCB vibration patterns carry important information about mechanical and electrical integrity and that vibration analysis requires a low level of invasiveness, this article presents methods of obtaining mechanism and reaction times using interference signals of outdoor high voltage SF6 circuit breakers (HVCB). Compared to traditional methods of monitoring outdoor SF6 circuit breakers that are based on encoders, the proposed method presented ease of installation, as it only requires the insertion of accelerometers. The method of obtaining the mechanism time is based on the use of interference and auxiliary contact signals, in the time domain, where the location of the accelerometer installation, in the structure of the HVCB, makes it possible to guarantee the moment of the trip command. To obtain the reaction time of each HVCB pole, the envelope technique was applied with the Hilbert transform module of the hearing signal, filtered in a certain resonance band. The proof of the technique of analyzing the vibration of the signal in time was developed with laboratory tests of an HVCB, instrumented with accelerometers and angular encoders. The results obtained by vibration analysis were compared with those obtained by angular encoders and it was concluded that with the acceleration signals, in time, it is possible to obtain performance parameters of an HVCB from its displacement curve. Finally, the online monitoring of the circuit breaker applied in the field is presented, where the acquisition of trip current signals, the condition of the SF6 gas and extinguishing current signals, were added to the instrumentation.

  1. Conclusions

Condition-based maintenance on outdoor high voltage SF6 circuit breakers (HVCB) is an important issue for the future of Smart Grids. For the evaluation of the condition of an HVCB, mechanism, insulation and contacts, the measurement of time and kinetic parameters by means of the travel curve is a competent method. However, access to the main contacts of each circuit breaker pole is inapplicable for online evaluation. Thus, this work presented a solution to obtain the parameters necessary for online monitoring in a non-invasive way. The method is based on obtaining the mechanism time and the reaction time of each pole of the HVCB using only acceleration and auxiliary contact signals. Experiments were conducted in an HVCB test laboratory, instrumented with accelerometers and angular encoders, and the results obtained with the vibration analysis, compared with those obtained by angular encoders, pointed to the effectiveness of the method in obtaining temporal performance parameters of a HVCB from its travel curve (mechanism and reaction time). In this work, the signals are processed in the time domain, which makes its automation simple, since it only depends on a function to detect the first peak of a signal obtained by the accelerometer. Compared to traditional methods of monitoring outdoor SF6 circuit breakers that are based on encoders, the proposed method presented ease of installation, as it only requires the insertion of accelerometers. The implementation of the field monitoring system was presented, containing additional instrumentation for the acquisition of trip current signals, the SF6 gas condition and extinguishing current signals. The system implemented in the field did not require structural changes to the HVCB, such as the installation of external panels and the passage of cables through additional ducts. Furthermore, the installed sensors did not require the synchronism panel to be de-energized or the terminals to be disconnected. Thus, it was possible to detect possible mechanical and/or electrical wear on the outdoor SF6 circuit breaker associated with degradation in the contact resistance of the poles, loss of the insulating characteristic of the SF6 gas and degradation of the mechanism.

Added in 1. Introduction:

Compared to the aforementioned methods, the proposed work performs the analysis in the time domain, as done in [10],[11],[12] and [13], which provides greater simplicity, as it is a direct analysis, without transformations. and of less complexity for the interpretation of the results, they are only one-dimensional arrays. This work, in comparison with the aforementioned time-domain monitoring techniques, has the advantage of not requiring the implementation of statistical tools, which reduces possible subjectivities in the interpretation of results, considering that it is the most direct method. Consequently, an algorithm for its automation becomes less complex.

Question 3

To comply with this item, it was added in 2.1. Event detection on the travel curve by acceleration signals:

It should be noted that the accelerometers were installed at the base of each circuit breaker pole, as for field installation, it is a safe location, without risk of electric shock and does not require the circuit breaker to be de-energized.

Question 4

To comply with this item, it was added in 4.2. Obtaining reaction time by vibration signal:

In order to elucidate the techniques developed in this work to obtain the HVCB mechanism and reaction time, Figure 12 below shows a flowchart with the mentioned stages of processing the acceleration signals.

Round 2

Reviewer 2 Report

After the revision of this new manuscript and the author’s responses; this reviewer can conclude the new submission has been significantly improved from the first version, and my concerns were clarified adequately.

Therefore, the new submitted article has been done well. The authors have addressed all issues.